# *Botryocladia leptopoda* Extracts Promote Wound Healing Ability via Antioxidant and Anti-Inflammatory Activities and Regulation of MMP/TIMP Expression

**DOI:** 10.3390/md23110444

**Published:** 2025-11-19

**Authors:** Shin-Ping Lin, Tsung-Kai Yi, Yi-Feng Kao, Ming-Chieh Tu, Chen-Che Hsieh, Yu-Chieh Chou, Jheng-Jhe Lu, Shella Permatasari Santoso, Huey-Jine Chai, Kuan-Chen Cheng

**Affiliations:** 1School of Food Safety, Taipei Medical University, Taipei 11031, Taiwan; 2Ph.D. Program in Drug Discovery and Development Industry, Taipei Medical University, Taipei 11031, Taiwan; 3Research Center of Biomedical Device, Taipei Medical University, Taipei 11031, Taiwan; 4Seafood Technology Division, Fisheries Research Institute, Ministry of Agriculture, Keelung 20246, Taiwan; 5Department of Seafood Science, College of Hydrosphere, National Kaohsiung University of Science and Technology, Kaohsiung 81157, Taiwan; 6Institute of Biotechnology, National Taiwan University, Taipei 10617, Taiwan; 7Department of Chemical Engineering, Faculty of Engineering, Widya Mandala Surabaya Catholic University, Surabaya 60114, Indonesia; 8Chemical Engineering Master Program, Widya Mandala Surabaya Catholic University, Surabaya 60114, Indonesia; 9Collaborative Research Center for Zero Waste and Sustainability, Widya Mandala Surabaya Catholic University, Surabaya 60114, Indonesia; 10Institute of Food Science and Technology, National Taiwan University, Taipei 10617, Taiwan; 11Department of Medical Research, China Medical University Hospital, China Medical University, Taichung 40402, Taiwan; 12Department of Optometry, Asia University, Taichung 41354, Taiwan; 13Department of Food Science, Fu Jen Catholic University, New Taipei City 242062, Taiwan

**Keywords:** *Botryocladia leptopoda*, cell proliferation, antioxidant, anti-inflammation, wound healing

## Abstract

Wound healing is a complex process involving coordinated actions of multiple cell types. Therefore, when developing therapeutics to promote wound healing, it is essential to consider the synergistic contributions of various cells at different stages of the healing process. In this study, we evaluated the potential of different extracts of *Botryocladia leptopoda* as wound-healing agents by examining their effects on antioxidant activity, cytotoxicity, cell migration, anti-inflammatory properties, and expressions of specific biomarkers associated with wound healing. Results indicated that the ethanol extract (FE) and hexane extract (HE) exhibited the highest DPPH radical scavenging activity, reaching up to 94%. The alkaline extract (AE) showed the strongest antioxidant ability in the FICA assay, with a maximum of 99%. In addition, the FE and AE provided anti-inflammatory actions that inhibited tumor necrosis factor (TNF)-α and interleukin (IL)-6 in lipopolysaccharide (LPS)-treated RAW 264.7 cells. Further analyses suggested that the FE and AE enhanced cell proliferation (210% and 112%) and migration (442.2% and 535.6%) and regulated wound healing-related genes, including matrix metalloproteinase 2, *MMP9*, and tissue inhibitor of metalloproteinase 2 (*TIMP2*) to avoid scar formation and accelerate wound healing. Lastly, the identification of potential compounds within the extract using the UHPLC system further supports its prospective medical applications. Taken together, these findings indicated that the FE and AE from *B. leptopoda* exhibited remarkable in vitro wound-healing properties, highlighting their potential for applications in pharmaceutical industries and health food development.

## 1. Introduction

Wound healing is an intricate process with interactions among various cells, extracellular compositions, and structures [1,2]. Different types of wounds require different care needs, such as accelerating wound healing, preventing pathogen infections, decreasing pain, and avoiding excessive tissue fluid leakage from the wound [3]. Although medications have been combined with dressings for wound care for many years, these medications often have side effects [4]. Natural compounds from marine or plant sources generally show better biocompatibility and lower cytotoxicity than synthetic drugs. They promote wound healing through mild antioxidant and anti-inflammatory actions, thereby reducing the risk of irritation or allergic reactions [5]. Therefore, developing novel wound-healing agents from natural sources has become a popular focus in recent years.

Marine algae are an unexploited reservoir of biologically active substances due to their abundant biodiversity [6]. Numerous studies have investigated their chemical constituents and bioactivities; however, further exploration of specific species and targeted applications remains highly valuable for future research.

Recent studies demonstrated that extracts from algae provide specific biological functions such as antioxidation [7], antimicrobial properties [8], and antiviral abilities [9], and could be helpful for issues such as high blood pressure [10], obesity problems [11], cardiac diseases [12], and specific cancers [13]. These biofunctions may arise from various types of bioactive molecules, such as proteins, compounds, polysaccharides, and peptides [14,15]. Therefore, it is essential to evaluate the functions and health benefits of different crude extracts within specific disease models, with the ultimate goal of isolating a purified bioactive molecule in alga-related research.

When applying marine algae for wound healing, most studies focused on developing wet wound dressings using algal polysaccharides [16,17], which form the structure of dressing materials and provide a surface for cell adhesion or attachment to promote wound healing. In addition, bioactive compounds from marine algae also participate in the wound-healing process and skin tissue regeneration by regulating cellular signaling pathways [18]. Given the high potential for the use of marine algae to promote wound healing, a number of ongoing studies are attempting to identify and isolate ideal materials for wound healing, especially through various types and rare macroalgae as research targets [16,19,20].

*Botryocladia leptopoda* (*B. leptopoda*) is a kind of marine red alga that has been used as feed, fertilizer, and in food manufacturing at an industrial level [21]. However, its biomedical potential, especially in skin repair and wound healing, remains largely unexplored. To the best of our knowledge, this is the first study to investigate the wound- healing potential of *B. leptopoda* extracts, providing novel insights into the therapeutic applications of marine red algae and expanding their value beyond conventional uses.

In our previous study [22], we focused on the effects of *B*. *leptopoda* extracts on the repair of skin cells under photo-damage conditions and found that the extracts exhibited strong protective activity. Therefore, the antioxidant and anti-inflammatory properties were further investigated. The cell-based experiments were conducted to assess cytotoxicity, cell adhesion, and cell proliferation for evaluating their potential for wound-care applications. Finally, the potential compounds present in the *B*. *leptopoda* extract were also analyzed using the negative ion mode of the ultra-high performance liquid chromatography (UHPLC) system to identify key bioactive constituents.

## 2. Results

### 2.1. Total Phenolic Content (TPC) and Antioxidant Capacity of B. leptopoda Extracts

In the past, many plants have been used for wound care and were later found to be rich in polyphenolic compounds [23]. These polyphenols exhibit strong anti-inflammatory effects [24], reduce oxidative stress [25], promote collagen regeneration [26], enhance re-epithelialization [27], and effectively accelerate wound healing [28]. Therefore, analyzing the TPC can aid in evaluating the potential of extracts from *B. leptopoda* as wound-care agents. *B*. *leptopoda* was extracted using four different methods: alkaline extraction with NaOH at 25 °C, ethanol extraction at 25 °C, ethanol extraction at 70 °C, and hot water extraction at 100 °C. The resulting products were designated as AE, FE, HE, and HW, respectively, according to their extraction processes.

Results (Figure 1a) indicated that phenolic compounds from *B. leptopoda* extracts were much more soluble in alkaline solvents than in water or weak polar organic solvents. The AE presented the highest TPC of 6.4 mg GA/g, compared to FE, HE, and HW (at 3.1, 2.7, and 1.7 mg/GA/g, respectively). Sun et al. [29] compared water, acidic, and alkaline solvent extracts and found the highest bioactive components were achieved in an alkaline environment. This may have been due to alkaline extraction helping break down algal cell walls composed of cellulose leading to release of components out of the algae [30]. Results from Peasura et al. [31] also showed a similar trend of alkaline extraction yielding higher sulfated polysaccharides compared to water and acidic extraction, which provided higher antioxidation abilities. Additionally, AE exhibited the highest polyphenol content, suggesting its potential as a promising wound-healing agent. Utpal et al. [32] have emphasized that polyphenols can modulate multiple stages of the wound-healing cascade, including antioxidation, inflammation control, angiogenesis, and tissue remodeling. Their bioactivities are primarily attributed to their strong radical-scavenging ability, regulation of cytokines (e.g., TNF-α, IL-6, VEGF, TGF-β1), and promotion of fibroblast and keratinocyte proliferation.

In results of the extraction efficiency (Figure 1b), the HW group showed the highest yield, reaching 41.3%. Concentrations in descending order were the AE group (21.1%), HE group (9.5%), and FE group (6.4%). Harb et al. [33] found that water extraction showed 3–13-fold higher yields compared to methanolic extraction, which was similar to our results. Owing to high polysaccharide contents in algae, the water extraction method was used to facilitate polysaccharide extraction, which may explain the results.

Free radical-scavenging properties were proven to promote wound-healing processes [34]. Therefore, evaluating the antioxidative abilities is crucial for biological ingredients applied to wound care, especially specific wounds such as diabetic wounds [35]. Results for DPPH (Figure 2a) showed that both the FE and HE groups exhibited strong antioxidation at 73.8~94.8% and 45.7~94.7%, respectively. These results suggested that the FE and HE samples contained significantly higher amounts of lipophilic antioxidants due to extraction with ethanol. In the AE and HW groups, hydrophilic solvents proved inadequate to extract lipophilic antioxidants, thereby resulting in reduced DPPH radical-scavenging activities (of 21.1~25.4% and 26.3~43.3%, respectively).

In the results of ferrous iron-chelating activity (Figure 2b), the AE group exhibited the highest ferrous iron-chelating capacity compared to the other groups. Interestingly, the iron-chelating activity followed a trend similar to that observed for TPC. This suggested that phenolic compounds in the AE may contribute to its high metal-chelating capacity. Previous studies showed that specific polyphenols derived from macroalgae possess ferrous iron-chelating properties [36,37]. Although the AE group demonstrated saturated ferrous iron-chelating activity, this finding confirmed its strong antioxidant potential. Subsequent experiments further assessed its wound-healing efficacy using different extract doses in cellular assays. Based on the potential of AE and FE in demonstrating antioxidant properties, these two groups were selected as the primary experimental groups for subsequent experiments.

### 2.2. Anti-Inflammatory Activity of B. leptopoda Extracts in Macrophages

In the non LPS-induced inflammation of TNF-α expression (Figure 3a), the TNF-α RNA level was found to have slightly increased or remained unchanged with increasing FE and AE treatment concentrations. Compared to the untreated group (92%), the highest treatment concentrations (1 mg/mL) reached 136% (in the FE group) and 81.75% (in the AE group). However, in the LPS-induced inflammation group, expression levels of TNF-α decreased from 13,376% to 6006.5% and 9072.2%, respectively.

Interestingly, a similar trend was also observed in the analysis of IL-6 (Figure 3b). In the LPS-induced inflammation group, the highest treatment concentrations of FE and AE (1 mg/mL) suppressed IL-6 expression from 4042.9% to 1811.5% and 2327%, respectively. These results indicated that both the FE and AE exhibited anti-inflammatory effects by inhibiting TNF-α and IL-6 expressions. These proinflammatory cytokines are involved in different types of inflammation, which lead to apoptosis and cell death at the wound. Apoptosis of dermal keratinocytes was considered to have delayed the wound-healing process [38]. Furthermore, the over-inflammation of a wound also causes release of reactive oxygen species (ROS) resulting in the formation of granulation tissues [39]. Therefore, the anti-inflammatory properties of an extract can help the wound-healing processes of proliferation and remodeling.

### 2.3. Cell Cytotoxicity, Proliferation, and Migration of Fibroblast Cells

In the cytotoxicity results (Figure 4), the FE and AE groups both maintained at least 80% cell viability at treatment concentrations below 1.5 mg/mL. However, at 2 mg/mL, cell viability significantly decreased to 70.59% and 80.3%, respectively. Therefore, test concentrations of <2 mg/mL were utilized in subsequent cell-based experiments.

In the late stage of wound healing such as proliferation and remodeling, fibroblasts will adhere and migrate to the wound to release various signaling factors to enhance angiogenesis and granulation tissue formation [40]. Hence, proliferation and migration abilities are crucial to evaluate the potential of wound-healing agents and dressings [41].

In Figure 5a, the FE group demonstrated a similar increase in proliferation effects following treatment with concentrations ranging from 0.25 to 1.5 mg/mL; however, a significant reduction in proliferation was observed at a concentration of 2 mg/mL. With AE treatment (Figure 5b), although treatment with concentrations exceeding 1.5 mg/mL showed a slight reduction in their ability to enhance the proliferation ability, the overall treatment effects still promoted cell proliferation, ranging from approximately 112.3% to 87.7%. The observed results may be associated with cytotoxicity. By comparing cytotoxicity results with proliferation outcomes, it was evident that treatment groups showing a decline in the differentiation capacity exhibited a significant upward trend in cytotoxicity. Therefore, lower treatment concentrations of extracts (0.25 and 0.5 mg/mL) were used in the migration experiment. In addition, the observation that lower concentrations of the extract promoted cell migration may also have resulted from other potential mechanisms such as the involvement of bioactive compounds exhibiting biphasic effects. For instance, endostatin is known to enhance cell migration at specific concentrations, yet inhibits this process at higher concentrations, reflecting its dual functional characteristics [42]. To confirm these effects, it is essential to isolate and characterize the active compounds for targeted validation through subsequent studies.

In Figure 6a, all treatment groups exhibited significant cell migration, with cells moving from the seeding area to the central void within 24 h. Interestingly, in both the FE and AE groups, 0.25 mg/mL treatment (442.2% and 535.6%, respectively) demonstrated superior migration performance compared to 0.5 mg/mL treatment (398.7% and 387.8%, respectively) (Figure 6b).

### 2.4. Evaluation of Scar Inhibition In Vitro

To evaluate the effect of *B. leptopoda* extracts on scar inhibition in vitro, a PMA-induced L929 cell model was utilized to investigate the RNA-level overexpression of MMPs, specifically *MMP2*, and *MMP9* (Figure 7a–c).

MMP2 is a kind of gelatinase which is involved in matrix remodeling, but its overexpression was implicated in delayed wound healing and tissue degradation [43]. MMP9 is essential for maintaining the dynamic equilibrium of the extracellular matrix (ECM) and is a key participant in wound-healing processes, serving as an important biomarker for anti-aging and skin regeneration [44]. Nevertheless, aberrant upregulation of MMP9 was associated with excessive granulation tissue formation [39], impaired healing, and the potential development of keloids [45].

In results of *MMP9* RNA expression (Figure 7c), the AE group exhibited a decrease in RNA expression from 199% to 136%. Interestingly, results for MMP2 (Figure 7a) showed that both the FE and AE inhibited *MMP2* RNA expression from 540% to 277% and 93%, respectively. These results suggested that the FE can inhibit MMP2 overexpression to avoid skin elasticity decrease and wrinkle formation. Jung et al. [46] reported that matrine can inhibit PMA-induced *MMP-1* mRNA expression by utilizing inhibition of activating protein (AP)-1 promoter activation, which may be helpful for anti-inflammation of dermatitis. Results also showed that the AE greatly repressed MMP2 and MM9, which may lead to acceleration of wound healing and wrinkle formation caused by collagen degradation.

However, this still needs to be further proven utilizing animal experiments. In addition, *TIMP-2* (the repressor gene of MMP2) RNA expression was also investigated (Figure 7c). Results indicated that AE treatment strongly enhanced TIMP-2 expression, which suggests that the AE can downregulate expression levels of MMP2 by modulating TIMP-2 activity. A previous study also demonstrated similar results that the *Davallia bilabiate* water extract decreased PMA-induced MMP2 expression by upregulating the *TIMP-2* RNA level, which provided a potential anti-angiogenic effect [47]. It should be noted that the current study evaluated MMP and TIMP expression only at the mRNA level, and it may still occur in the correlation of error with protein expression; therefore, future studies will therefore include protein-level validation and enzymatic assays to confirm the regulatory effects of the extracts on the MMP/TIMP pathway.

In this study, based on the high TPC of AE and FE, the negative ion mode of ultra-high-performance liquid chromatography (UHPLC) was used to characterize these two extracts (Table 1 and Appendix A). Both were found to primarily exhibit functions associated with anti-inflammatory, antioxidant, and immunomodulatory activities. However, variations in the proportional distribution of these components were observed, which may account for the comparable effects of AE and FE in these functional domains. These findings provide additional evidence supporting their potential to promote wound healing. In addition, although UHPLC-MS was employed in this study for preliminary component identification, this analysis was insufficient to confirm the active constituents. Therefore, further studies will perform advanced characterization to determine the key bioactive compounds and elucidate their structures.

## 3. Discussion

The present study demonstrated that AE and FE from *B. leptopoda* exhibit significant antioxidant, anti-inflammatory, and wound-healing properties. These biological activities can be attributed to the nitrogen-containing heterocycles and phenolic derivatives identified in the extracts. In agreement with previous studies also showing that phenolic and benzoic acid derivatives from marine algae exert potent anti-inflammatory and antioxidant effects that promote tissue regeneration [23,48,49], the high TPC observed in the AE supports its strong bioactive potential. AE group not only yielded higher TPC but also enhanced the solubilization of cell wall-bound phenolics by partially disrupting the cellulose structure of algal cell walls, consistent with prior findings by Sun et al. [29]. Although most of the compounds identified in the AE and FE fractions were not polyphenolic in nature, the TPC assay used in this study still exhibited high apparent TPC values due to the strong reducing capacity of these compounds. For instance, indole derivatives possess mild reducing power, whereas 4-hydroxybenzaldehyde contains a phenolic hydroxyl group contributing to the reaction.

The antioxidant assay results revealed that FE exhibited higher free radical scavenging activity, indicating the presence of lipophilic antioxidant compounds effectively extracted by ethanol. In contrast, AE showed a stronger ferrous ion-chelating capacity, consistent with its higher TPC value, suggesting a strong correlation between the concentration of extracted compounds and their metal-chelating activity. These findings are consistent with previous evidence that marine-derived indole and phenolic compounds can scavenge free radicals and bind transition metals, thereby alleviating oxidative stress and facilitating the wound-healing process [50,51].

The anti-inflammatory activity of the AE and FE were found to suppress TNF-α and IL-6 mRNA expression in LPS-induced macrophages. Overexpression of these proinflammatory cytokines is known to delay cellular regeneration by promoting apoptosis and excessive ROS generation. Therefore, the downregulation of TNF-α and IL-6 suggested that the extracts decreased the inflammatory cascades and restore redox balance to enhance the cell proliferation and remodeling [38,39].

In vitro scar inhibition assays revealed that both AE and FE could inhibit PMA-induced overexpression of MMP2, and MMP9 gene. These genes were related to the degradation of collagen and extracellular matrix components, leading to delayed healing and wrinkle formation [43]. Especially, AE significantly reduced MMP2 and MMP9 expression while enhancing TIMP-2 level, implying its regulatory effect on the MMP/TIMP. This balance is essential for maintaining extracellular matrix homeostasis and preventing excessive tissue remodeling [45]. This regulation may explain the superior wound-healing observed for AE in cell model.

UHPLC analysis identified quinolin-3-ol as the predominant compound in FE group, whereas indoleacetic acid was the major constituent in the AE group. Quinolin-derivatives have been reported to possess strong antioxidant and anti-inflammatory activities [52], which could explain the superior ROS-scavenging capacity and cytokine inhibition observed in FE-treated groups. In contrast, the study of indoleacetic acid and wound healing remains limited. Some studies have explored its use as a hydrogel component to enhance wound repair [53]; however, its underlying mechanism of action has not yet been clearly elucidated. Therefore, it still requires further identification and isolation to confirm the actual active components present in the extract.

In addition, the biochemical composition of macroalgae varies with species, season, and environmental conditions, which can influence their bioactivities. Although a detailed composition analysis of *B. leptopoda* was not performed in this study, previous reports indicate that *Botryocladia* species are rich in phenolic compounds with antioxidant and anti-inflammatory potential [21]. Future work will focus on correlating these constituents with the observed wound-healing effects.

In summary, these findings suggest that AE and FE from *B. leptopoda* exhibit multiple therapeutic functions—combining antioxidant and anti-inflammatory activities, as well as promoting cell proliferation and cell adhesion. These functions may contribute to skin regeneration and scar inhibition. The identification of key compounds not only reveals their biological mechanisms but also highlights the potential of *B. leptopoda* extracts for pharmaceutical and cosmeceutical applications. In future work, in vivo studies are necessary to further validate their potential for developing wound-healing formulations.

## 4. Materials and Methods

### 4.1. Materials and Chemicals

RAW 264.7 macrophage and L929 fibroblast cell lines were obtained from American Type Culture Collection (ATCC; Manassas, VA, USA). EDTA, trypsin, and antibiotics (penicillin/streptomycin) were obtained from GE Healthcare Life Science (Logan, UT, USA). Vivaspin 15R and Vivaspin Turbo 15 were obtained from Sartorius Stedim Biotech (Goettingen, Germany). Cell culture media (Dulbecco’s modified Eagle medium (DMEM), supplemented with high glucose, phenol red, sodium pyruvate, and L-glutamine) and fetal bovine serum (FBS) were procured from Hyclone Laboratories (Logan, UT, USA). All chemicals used in this study were of analytical grade and purchased from Merck (Burlington, MA, USA). The Cell Proliferation ELISA (enzyme-linked immunosorbent assay) BrdU kit was purchased from Roche Applied Science (Mannheim, Germany). MTS reagent powder (G1111) and phenazine methosulfate were purchased from GeneLabs Life Science Corp. (Taipei, Taiwan).

### 4.2. Extraction of B. leptopoda

Raw material of *B. leptopoda* was obtained from the Eastern Marine Fishery Research Center of the Ministry of Agriculture (Taitung, Taiwan). After being thoroughly washed, the alga was dried and pulverized. The powder was sieved through a 40-mesh screen. *B. leptopoda* powder was then extracted at 25 °C using sodium hydroxide (alkaline extract, AE), at 25 °C using ethanol (FE), at 70 °C using ethanol (HE), and at 100 °C using deionization and distilled (dd)H_2_O (HW extract). Extraction was performed using different solvents and temperatures under ultrasonic treatment at 20 kHz in an ice bath for 30 min. The mixtures were then centrifuged at 13,000× *g*, and the supernatants were collected and stored in light-protected tubes at −20 °C for further experiments. All experiments were conducted as repeated measurements using the same extraction sample, while independently re-extracted samples were also examined to verify the reproducibility and consistency of the results.

### 4.3. Identification of B. leptopoda Extracts Using Ultra-High-Performance Liquid Chromatograph (UHPLC)

In this study, 10 mg of freeze-dried microalgal biomass was suspended in 1 mL of chromatography-grade methanol containing 1% BHT, vortexed for 5 min, and then stored at 4 °C for 1 h. The mixture was subsequently subjected to ultrasonic extraction at 20 kHz in an ice bath for 30 min, followed by centrifugation at 12,000× *g* for 5 min at 4 °C. The resulting supernatant was filtered through a 0.22 µm PTFE membrane and used for further analysis. The chemical composition of the extract was analyzed using an UHPLC system (Vanquish Horizon, Thermo Scientific, Milan, Italy) coupled with a quadrupole-Orbitrap high-resolution mass spectrometer (Q-Orbitrap-HRMS) (Thermo Scientific). Separation was achieved on a Syncronis C18 column under chromatographic conditions, with a flow rate of 0.3 mL·min^−1^ and an injection volume of 5.0 µL. Full-scan mass spectrometric data were acquired in negative ion modes at a resolving power of 70,000 FWHM, covering an *m*/*z* range of 100–1200. The automatic gain control (AGC) target was set to 5 × 10^5^ ions, with a maximum injection time of 200 ms. Data-dependent acquisition (DDA) was applied to obtain product ion spectra using collision energies of 45–60 eV and a resolution of 35,000 FWHM. A mass inclusion list containing precursor ion *m*/*z* values of target compounds and acquisition windows was employed.

### 4.4. Total Phenolic Content (TPC)

The TPC was used according to a method described by Pérez et al. [54] with slight modification: 0.25 mL of Folin–Ciocalteu reagent was mixed with 0.25 mL of an extract (volume ratio of 1:1), and the mixture was thoroughly mixed for 3 min at room temperature. Fifty milliliters of a 20% Na_2_CO_3_ solution was slowly added and incubated at 40 °C for 20 min. The absorbance of the mixture was determined at 755 nm. The TPC was calculated using a gallic acid standard curve, and results are expressed as milligrams of gallic acid equivalents per milliliter (mg GAE/mL).

### 4.5. Antioxidation Ability

Different concentrations of extracts (2, 5 and 10 mg/mL) were prepared using 80% methanol, and were shaken and extracted at 150 rpm and 25 °C for 30 min, followed by centrifugation at 8000× *g* and 4 °C for 20 min, and the supernatant was collected and used in the following experiment. L-Ascorbic acid (2, 5, and 10 mg/mL) was used as the positive control in the antioxidation ability assays.

In the diphenyl-1-picrylhydrazyl (DPPH) free radical-scavenging activity assay [55], 0.1 mM methanolic DPPH as a reaction solution was mixed with an extract (volume ratio of 1:1) at 25 °C in the dark for 30 min, and the absorbance value at 517 nm was measured.

The DPPH assay is based on the reduction in the stable purple DPPH radical to a yellow-colored diphenylpicrylhydrazine upon reaction with an antioxidant, which donates hydrogen or electrons. The decrease in absorbance at 517 nm indicates the free radical-scavenging ability of the sample.

Results were expressed as the scavenging activity (%) using Equation (1):(1)$$Scavenging\;activity\;\left(\%\right)=\frac{A_{nc}-\left(A_{sample}-A_{blank}\right)}{A_{nc}}\times 100\%;$$
where A*_nc_*, A*_sample_*, and A*_blank_* are the absorbance at 517 nm for the negative control (0.1 mM methanolic DPPH, without the addition of sample), sample (0.1 mM methanolic DPPH, with the addition of extract at a specific concentration), and blank (sample with ddH_2_O), respectively. Trolox (6-hydroxy-2,5,7,8-tetramethyl chroman-2-carboxylic acid) was used as the standard antioxidant for the assay.

In the ferrous ion-chelating activity (FICA) assay [56], 1 mL of extract was mixed with 0.1 mL of a 2 mM FeCl_2_ solution for 30 s, and reacted with 0.2 mL of 5 mM ferrous hydrazine at room temperature for 10 min. The absorbance value at 562 nm was measured.

The FICA assay is based on the ability of antioxidants to compete with ferrozine for ferrous ions (Fe^2+^). When chelating agents are present, they form complexes with Fe^2+^, reducing the formation of the red-colored ferrozine–Fe^2+^ complex, which results in a lower absorbance at 562 nm.

Results are expressed as ferrous ion-chelating activity (%) using Equation (2):(2)$$Ferrous\;ion-chelating\;activity\left(\%\right)=\frac{A_{nc}-\left(A_{sample}-A_{blank}\right)}{A_{nc}}\times 100\%;$$
where A*_nc_*, A*_sample_*, and A*_blank_* are the absorbance at 562 nm for the negative control (2 mM FeCl_2_ solution, with ddH_2_O), sample (2 mM FeCl_2_ solution, with the addition of extract at specific concentration), and blank (sample with ddH_2_O), respectively.

### 4.6. Detection of Anti-Inflammatory Ability

To confirm the ability of inflammatory inhibition by *B. leptopod* extracts, lipopolysaccharide (LPS)-treated RAW 264.7 macrophages were used as the detection cell model.

The anti-inflammatory was evaluated using ELISA kits to analyze levels of interleukin (IL)-6 (DY 406, R&D Systems Inc., Minneapolis, MN, USA), and tumor necrosis factor (TNF)-α (DY 410, R&D Systems Inc., Minneapolis, MN, USA). A 96-well plate coated with capture antibodies of IL-6 and TNF-α was washed with washing buffer, and 1× assay diluent (200 μL/well) was added at room temperature for 1 h. The assay diluent, AE or FE (62.5, 125, 250, 500, and 1000 µg/mL), and RAW 264.7 macrophage cells (10^6^ cells/mL) with or without LPS treatment (1 µg/mL) were mixed at room temperature for 1 h. After washing buffer treatment, avidin-horseradish peroxidase (HRP) (100 μL/well) was added, and incubated for 30 min. After washing buffer treatment, the 3,3,5,5′-tetramethylbenzidine (TMB) substrate (100 μL/well) was added and incubated for 15 min in a dark room. The reaction was stopped by adding a stop solution (50 μL/well), and the OD value at 450 nm was detected.

### 4.7. Cell Cytotoxicity, Proliferation, and Migration

For the cell cytotoxicity assay, L929 fibroblast cells were cultured in 90% DMEM and 10% FBS for 3 days in an incubator at 37 °C with 5% CO_2_ airflow. L929 fibroblasts at a concentration of 10^6^ cell/mL were plated in a 96-well plate with culture medium containing different concentrations of *B. leptopoda* extracts for 4 h of incubation. Treated cells were washed twice with phosphate-buffered saline (PBS), then 120 μL of 10-fold diluted MTS reagent was added to each well. Cells were left for 1 h in an incubator at 37 °C in the dark. The number of viable cells was measured at a wavelength of 490 nm. The diluted MTS reagent was used as a blank, and wells containing cells without extract addition were used as a negative control. Cell viability was calculated by comparing absorbance values of treated groups to that of the control group, which was set to 100%.

In the cell proliferation assay, L929 fibroblasts at a concentration of 2 × 10^5^ cells/mL were plated in a 96-well plate with culture medium containing different concentrations of *B. leptopoda* extracts for 24 h of incubation and mixed with a BrdU solution (10 μL/well, 10 μM) for 4 h at 37 °C. After the medium was removed, cells were fixed and incubated for 30 min at room temperature. A solution of a BrdU antibody conjugated with peroxidase was added (100 μL/well) for 90 min of treatment. Wells were washed with washing solution (200 μL/well), and a substrate solution (100 μL/well) was added. The absorbance was monitored at 370 nm. The proliferation ability was calculated by comparing absorbance values of treated groups to that of the control group, which was set to 100%.

In the cell-migration assay, L929 cells were inoculated in a culture dish with a central void and incubated until 80~90% cell confluency was reached. Cells migrated directly from the surroundings into the central void, and cell fragments were cleaned with PBS. Different concentrations of extracts were added to the serum-free medium. The central void was measured under an inverted microscope at 24 h. ImageJ software (1.53 k, National Institutes of Health, Bethesda, MD, USA) was used to calculate the area of the void area, and the migration rate was calculated using Equation (3):(3)$$Migration\;rate\left(\%\right)=\frac{A_{0}-A_{24}}{A_{0}}\times 100\%;$$
where A*_0_* and A*_24_* are areas of the void area at 0 and 24 h, respectively, after treatment with the extract.

### 4.8. Quantitative Real-Time Polymerase Chain Reaction (qRT-PCR)

To detect the anti-scarring effects of different *B. leptopoda* extracts, messenger (m)RNA expressions of genes including matrix metalloproteinase genes (*MMP-1*, *MMP-2*, and *MMP-9*) and tissue inhibitor of metalloproteinase 2 (*TIMP-2*) were investigated. L929 cells (6 × 10^5^ cell/mL) were seeded inT25 flasks for 4 h. Medium with extract was added and incubated at 37 °C for 24 h. Phorbol-12-myristate 13-acetate (PMA) (1 μg/mL) was also added to the medium before incubation to induce MMP-related gene activation.

Total RNA extracted from cell samples was converted to complementary (c)DNA using an iScript cDNA Synthesis Kit (Bio-Rad, Hercules, CA, USA). A reverse-transcription quantitative polymerase chain reaction (RT-qPCR) analysis was carried out using iScript RT-qPCR Sample Preparation Reagent (Bio-Rad).

The qPCR conditions were 2 min at 50 °C and then 10 min at 95 °C. Afterward, conditions were set to 40 cycles of 15 s in which the temperature was kept at 95 °C and subsequently dropped to 60 °C for 60 s. The qPCR analysis involved the use of specific forward (F) and reverse (R) primers, namely MMP-2 RNA-F (5′-ATCGCAGACTCCTGGAATG-3′) and MMP-2 RNA-R (5′-CCAGCCAGTCTGATTTGATG-3′) for the *MMP-2* RNA gene, MMP-9 RNA-F (5′-GAGCCCTAGTTCAAGGGCAC-3′) and MMP-9 RNA-R (5′-TCCAGTACCAAGACAAAGC-3′) for the *MMP-9* RNA gene, and TIMP-2 RNA-F (5′-GGTAGCCTGTGAATGTTCCT-3′) and TIMP-2 RNA-R (5′-ACGAAAATGCCCTCAGAAG-3′) for the *TIMP-2* gene. Expression levels were standardized to those of the housekeeping gene, GAPDH RNA-F (5′-ACTCCACTCACGGCAAATTCA-3′) and GAPDH RNA-R (5′-CGCTCCTGGAAGATGGTGAT-3′) for the mouse *GAPDH* gene. Calculation formulas were as follows: Relative expression of gene = 2ΔΔCT.$$△CT=CT\left(target\;gene\right)-CT\;(reference\;gene)$$$$△△CT=△Ct\left(control\;target\;gene\right)-△Ct\;(treatment\;target\;gene)$$

### 4.9. Statistical Analysis

All experiments were independently repeated at least three times, and the obtained data are expressed as the mean ± standard deviation (SD). Statistical data analyses were performed using SPSS Statistics 22 (IBM, Armonk, NY, USA). A one-way analysis of variance (ANOVA) with Tukey’s honest significant difference (HSD) test was applied to determine significant differences (*p* < 0.05).

## 5. Conclusions

In summary, different *B. leptopoda* extracts were investigated for their in vitro antioxidation, proliferation, migration, and scar-inhibition activities. Result for the TPC showed that the AE had the highest phenolic compound contents. As to the antioxidative ability, the FE and HE presented the greatest DPPH-scavenging activities, and the AE exhibited an excellent FITA antioxidant ability. The FE and AE blocked LPS-induced inflammation in RAW 264.7 macrophages via suppressing proinflammatory cytokines such as TNF-α and IL-6 in concentration-dependent manners. As to proliferation, the FE and AE both presented an ability to enhance cell proliferation and migration at a low concentration of 0.25 mg/mL. Lastly, the FE and AE suppressed MMP2 overexpression, while further, respectively, inhibiting MMP9. This may help prevent wrinkle formation and the loss of skin elasticity and promote wound healing. In the future, further studies will focus on exploring the effects of specific compounds within the extracts on wound healing. Additionally, animal experiments will be conducted to validate the biological efficacy of these extracts in promoting wound healing.

## Figures and Tables

**Figure 1 marinedrugs-23-00444-f001:**
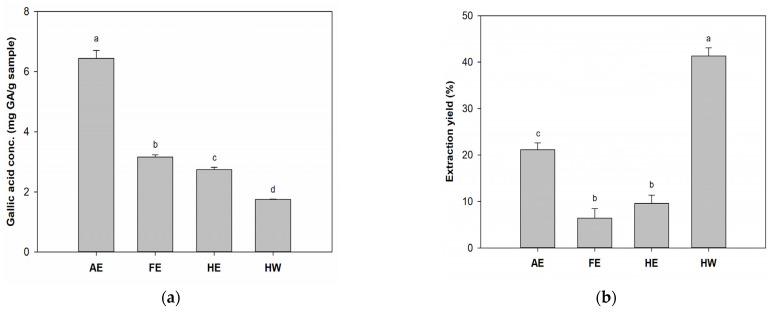
Effects of different solvent extracts on total phenolic contents of *Botryocladia leptopoda* (**a**, **b**) extraction yields. Different letters on the error bars indicate significant differences among the extraction groups and treatments (*p* < 0.05). AE: alkaline extract obtained at 25 °C; FE: ethanol extract obtained at 25 °C; HE: ethanol extract obtained at 70 °C; HW: hot water extract obtained at 100 °C.

**Figure 2 marinedrugs-23-00444-f002:**
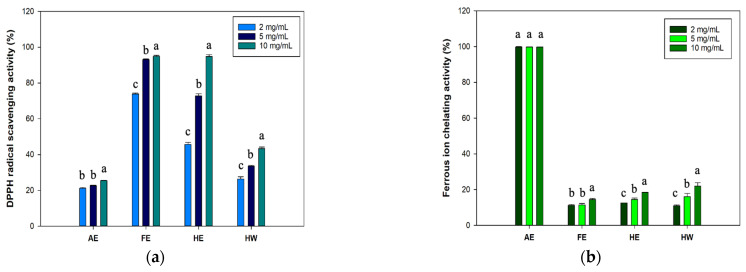
Antioxidant effects of various extracts (**a**) against DPPH radicals and (**b**) in a FICA assay. Different letters on the error bars indicate significant differences among the extraction groups and treatments (*p* < 0.05). AE: alkaline extract obtained at 25 °C; FE: ethanol extract obtained at 25 °C; HE: ethanol extract obtained at 70 °C; HW: hot water extract obtained at 100 °C.

**Figure 3 marinedrugs-23-00444-f003:**
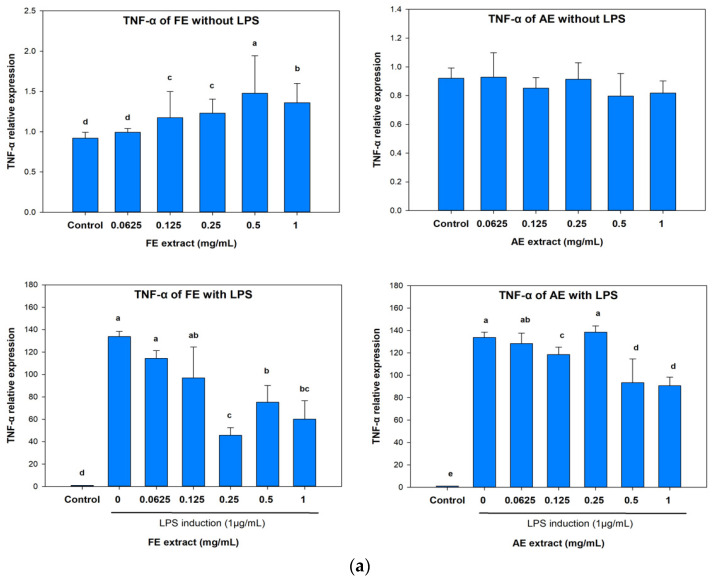
Effects of *Botryocladia leptopoda* extracts on expression levels of proinflammatory-associated genes in LPS-stimulated RAW 264.7 macrophages. (**a**) *TNF-α* and (**b**) *IL-6*. Different letters on the error bars indicate significant differences among the different concentrations of the extract (*p* < 0.05). AE: alkaline extract obtained at 25 °C; FE: ethanol extract obtained at 25 °C.

**Figure 4 marinedrugs-23-00444-f004:**
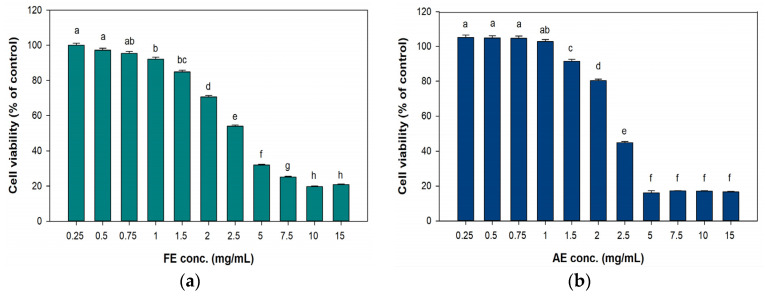
Cytotoxic effects on L929 fibroblast cells under different concentrations of (**a**) FE and (**b**) AE treatment. Different letters on the error bars indicate significant differences among the different concentrations of the extract (*p* < 0.05). FE: ethanol extract obtained at 25 °C; AE: alkaline extract obtained at 25 °C.

**Figure 5 marinedrugs-23-00444-f005:**
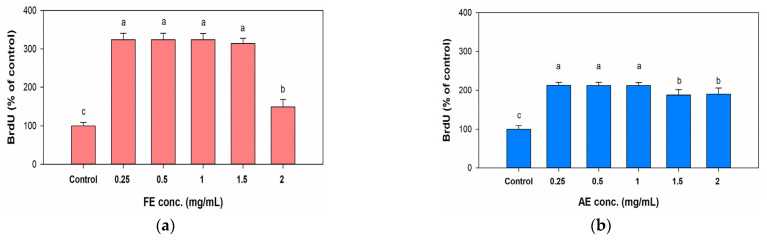
Cell proliferation on L929 fibroblast cells under different concentrations of (**a**) FE and (**b**) AE treatment. Different letters on the error bars indicate significant differences among the different concentrations of the extract (*p* < 0.05). FE: ethanol extract obtained at 25 °C; AE: alkaline extract obtained at 25 °C.

**Figure 6 marinedrugs-23-00444-f006:**
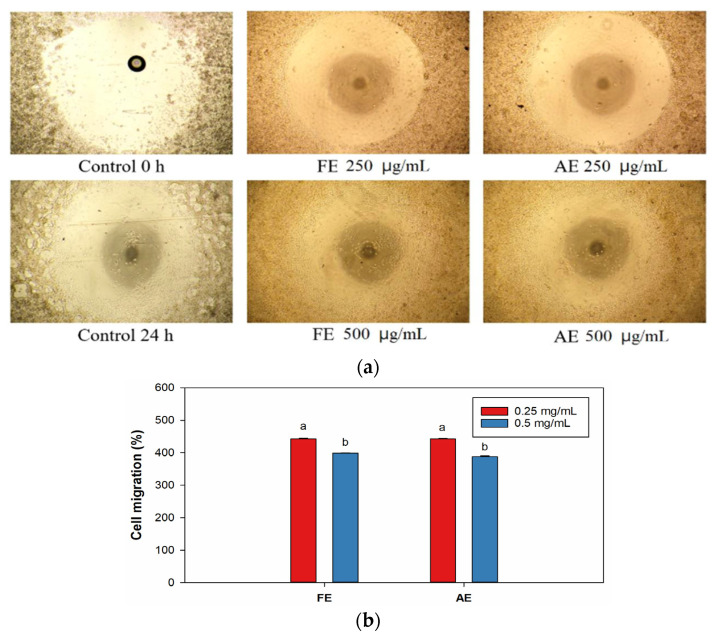
Cell migration of L929 cells under different concentrations of *Botryocladia leptopoda* extracts. (**a**) In vitro scratch assay of cell diffusion and (**b**) quantitative analysis of cell diffusion. Original magnification ×40. Different letters on the error bars indicate significant differences among the different concentrations of the extract (*p* < 0.05). AE: alkaline extract obtained at 25 °C; FE: ethanol extract obtained at 25 °C.

**Figure 7 marinedrugs-23-00444-f007:**
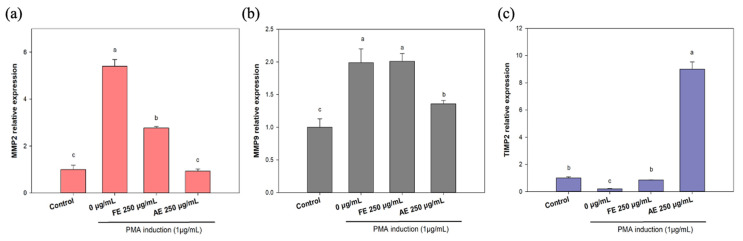
Effects of different *Botryocladia leptopoda* extracts on H_2_O_2_-induced changes in an intracellular reduced ratio of (**a**) *MMP-2*, (**b**) *MMP-9* and (**c**) *TIMP-2*, mRNA expressions. Different letters on the error bars indicate significant differences among the different concentrations of the extract (*p* < 0.05). AE: alkaline extract obtained at 25 °C; FE: ethanol extract obtained at 25 °C.

**Table 1 marinedrugs-23-00444-t001:** Composition analysis of *B. leptopoda* extracts in negative ion mode. AE: alkaline extract obtained at 25 °C; FE: ethanol extract obtained at 25 °C.

Proportion (%)	Compound	Potential Skin Care Related Function
FE		
31.25	Quinolin-3-ol	Preservatives, disinfectants and pesticides
10.8	8-Hydroxyquinoline-5-carboxylic acid	Antioxidant
10.05	6-Methylnicotinic acid	Anti-inflammatory
8.61	4-Hydroxybenzaldehyde	Preservatives, skin whitening
5.64	1H-Indole-3-carboxylic acid	Antimicrobial property
3.84	4-Hydroxyquinoline	Dehydrogenase inhibition
AE		
37.2	Indoleacetic acid	Skin moisturizing, anti-oxidation
21.23	Quinolin-3-ol	Preservatives, disinfectants and pesticides
8.17	1H-Indole-3-carboxylic acid	Antimicrobial property, anti-inflammatory
3.6	2-(Formylamino)benzoic acid	Preservatives

## Data Availability

The data presented in this study are available on request from the corresponding author.

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
