# Peer review of "Botryocladia leptopoda* Extracts Promote Wound Healing Ability via Antioxidant and Anti-Inflammatory Activities and Regulation of MMP/TIMP Expression"

_marinedrugs, 2025, doi:10.3390/md23110444_

Round 1
Reviewer 1 Report
Comments and Suggestions for Authors
The manuscript presents a relevant study on the biological activities of Botryocladia leptopoda extracts, focusing on antioxidant, anti-inflammatory, and in vitro wound healing properties. The topic is current, falls within the scope of the journal Marine Drugs, and has potential applicability in marine biotechnology and pharmacology of natural products.
However, the work still lacks further experimental depth and mechanistic validation. Some of the conclusions are supported only by descriptive analyses, and some results are not sufficient to fully support the claims made, especially regarding MMP/TIMP regulation and wound healing activity.
Below are the comments I made about the manuscript.
Major
Despite the use of UHPLC–MS, the identification of the compounds is only preliminary and based on exact masses, without structural confirmation. The authors should perform MS/MS analyses or compare them with authentic standards of the main compounds identified (e.g., indoleacetic acid, quinolin-3-ol). If this is not possible, the authors should make this limitation clear in the discussion and conclusion of the manuscript.
The discussion regarding MMP/TIMP regulation is based solely on mRNA-level data. The authors should perform assays with enzymes. For example, they could conduct Western blot or ELISA to measure the protein levels of MMP-2, MMP-9, and TIMP-2. Alternatively, they could perform gelatin zymography to assess the actual enzymatic activity of the MMPs.
Regarding the cell migration and proliferation assays. It can be said that the "scratch" test was applied more qualitatively than quantitatively. I suggest that the authors quantify the percentage of wound closure over time using ImageJ. Or they could perform a Transwell migration assay.
The authors should include a positive control, such as EGF or ascorbic acid, for comparison of effects.
The conclusions regarding the anti-inflammatory action were based on mRNA expression. The authors should complement this observation with measurements of TNF-α and IL-6 present in the cellular environment, using, for example, an ELISA technique. Or they should explicitly state this limitation of their data in the discussion and conclusion.
In conclusion, I think the term "wound healing" was misused. This event involves multiple cell types (fibroblasts, keratinocytes, and endothelial cells). The study only used fibroblasts and macrophages. I suggest that the authors conduct further emigration/proliferation experiments with the samples, but this time using a keratinocyte (e.g., HaCaT). Or they should modify the term used in the manuscript.
It is unclear which statistical method was used to compare the effect of the samples in the different trials. This information needs to be included in the methods section. Furthermore, in all the figure captions, the authors do not explain what the statistical markers presented in the figures mean.
Minor
The English in the text needs to be revised.
Figures: Some images (e.g., migration test) lack adequate scale and contrast bars.
Discussion: The authors should explore more critically the differences between the AE and FE extracts and correlate their chemical profiles with the observed biological activities
Comments on the Quality of English LanguageThe English in the text needs to be revised.
Author Response
- Despite the use of UHPLC–MS, the identification of the compounds is only preliminary and based on exact masses, without structural confirmation. The authors should perform MS/MS analyses or compare them with authentic standards of the main compounds identified (e.g., indoleacetic acid, quinolin-3-ol). If this is not possible, the authors should make this limitation clear in the discussion and conclusion of the manuscript.
[Response]: We thank the reviewer for pointing this out. Since this study serves as a preliminary verification of wound-healing efficacy, only crude extracts were subjected to initial analysis. Further studies will employ more advanced identification techniques to analyze the active components and conduct animal experiments to validate their wound-healing effects. We have added these limitations in the Discussion section.
“These findings provide additional evidence supporting their potential to promote wound healing. In addition, although UHPLC-MS was employed in this study for preliminary component identification, this analysis was insufficient to confirm the active constituents. Therefore, further studies will perform advanced characterization to determine the key bioactive compounds and elucidate their structures.”
- The discussion regarding MMP/TIMP regulation is based solely on mRNA-level data. The authors should perform assays with enzymes. For example, they could conduct Western blot or ELISA to measure the protein levels of MMP-2, MMP-9, and TIMP-2. Alternatively, they could perform gelatin zymography to assess the actual enzymatic activity of the MMPs.
[Response]: We appreciate the reviewer’s comment. We agree that the assessment of MMP/TIMP regulation at the mRNA level alone does not fully reflect their enzymatic activity. Due to the preliminary nature of this study, we focused on transcription level as an initial indication of wound-healing regulation. In future studies, we plan to perform protein-level analyses and enzymatic activity assessments of the related factors in a mouse wound-healing model.
- Regarding the cell migration and proliferation assays. It can be said that the "scratch" test was applied more qualitatively than quantitatively. I suggest that the authors quantify the percentage of wound closure over time using ImageJ. Or they could perform a Transwell migration assay.
[Response]: We thank reviewer for the suggestion. We agree that the scratch assay in the current study was performed qualitatively to demonstrate the wound-closure tendency. In future work, we plan to quantify the percentage of wound closure over time using ImageJ analysis to provide more objective evidence. Additionally, cell migration ability will be further validated using a Transwell migration assay in subsequent studies.
- The authors should include a positive control, such as EGF or ascorbic acid, for comparison of effects.
[Response]: Thank you for your suggestion. As this study focused on the preliminary evaluation of the wound-healing potential of the algal extracts, we were not yet certain about their healing efficacy or underlying mechanisms; therefore, no positive control group was included at this stage. Based on our current findings, the extracts appear to modulate MMP-related pathways. In future animal studies, we plan to include doxycycline or epidermal growth factor (EGF), as you suggested, as positive controls for comparison.
- The conclusions regarding the anti-inflammatory action were based on mRNA expression. The authors should complement this observation with measurements of TNF-α and IL-6 present in the cellular environment, using, for example, an ELISA technique. Or they should explicitly state this limitation of their data in the discussion and conclusion.
[Response]: Thank you for your comment. We agree that relying solely on RNA expression does not necessarily reflect corresponding protein-level changes. Therefore, we have noted this limitation in the Discussion section to inform readers and provide context for interpreting our findings.
“It should be noted that the current study evaluated MMP and TIMP expression only at the mRNA level. and it may still occur in the correlation of error with protein expression. therefore, future studies will therefore include protein-level validation and enzymatic assays to confirm the regulatory effects of the extracts on the MMP/TIMP pathway.”
- In conclusion, I think the term "wound healing" was misused. This event involves multiple cell types (fibroblasts, keratinocytes, and endothelial cells). The study only used fibroblasts and macrophages. I suggest that the authors conduct further emigration/proliferation experiments with the samples, but this time using a keratinocyte (e.g., HaCaT). Or they should modify the term used in the manuscript.
[Response]: We agree that using only cell-based models is not sufficient to comprehensively validate wound-healing efficacy. Thank you for pointing this out. Accordingly, we will revise the terminology in the manuscript to use “wound-healing ability” and “cellular regeneration” to more accurately describe the scope of this study.
- It is unclear which statistical method was used to compare the effect of the samples in the different trials. This information needs to be included in the methods section. Furthermore, in all the figure captions, the authors do not explain what the statistical markers presented in the figures mean.
[Response]: We thank you for the comment. all the explanation of statistical markers has been added.
Minor
- The English in the text needs to be revised.
[Response]: The English text has been revised and highlighted in red.
- Figures: Some images (e.g., migration test) lack adequate scale and contrast bars.
[Response]: We thank the reviewer for the suggestion. The migration test has been updated to include the corresponding magnification.
- Discussion: The authors should explore more critically the differences between the AE and FE extracts and correlate their chemical profiles with the observed biological activities
[Response]: We thank the reviewer for the suggestion. In the revised manuscript, we have added a discussion highlighting the chemical and functional differences between the FE and AE extracts.
. “UHPLC analysis identified quinolin-3-ol as the predominant compound in FE group, whereas indoleacetic acid was the major constituent in the AE group. Quinolin-derivatives have been reported to possess strong antioxidant and anti-inflammatory activities[46], which could explain the superior ROS-scavenging capacity and cytokine inhibition observed in FE-treated groups. In contrast, the study of indoleacetic acid and wound healing remains limited. Some studies have explored its use as a hydrogel component to enhance wound repair[47]; however, its underlying mechanism of action has not yet been clearly elucidated. Therefore, it still requires further identification and isolation to confirm the actual active components present in the extract.”

Reviewer 2 Report
Comments and Suggestions for Authors
Comments to the Authors
In the Manuscript titled “Botryocladia leptopoda Extracts Promote Wound Healing via Antioxidant and Anti-Inflammatory Activities and Regulation of MMP/TIMP Expression”, extracts (identity/composition of extracts AE, FE, HE, and HW unknown) from the marine alga B. leptopoda are used to determine potential use for wound healing through in vitro, cell-based experiments using mouse RAW 264.7 macrophage and L929 fibroblast cell lines (reason for not using similarly available human cell lines not given) and PMA as tumor promoter. To this end, a) antioxidant effects of different extract concentrations in cell-free assays are measured by colorimetric assays (Fig.2 - sample identification and statistical significance of results unknown), b) anti-inflammatory effects in RAW 264.7 and L929 cell lines are measured by ELISA assays of TNF-α and IL-6 (Fig.3 – reagents sources and statistical significance of results unknow), c) cytotoxicity (Fig.4 - MTS assay, statistical significance unknown), proliferation (Fig.5 - BrdU incorporation ELISA, statistical significance unknown), and cell migration (Fig.6 - microscopy, ImageJ software, statistical significance unknown) are determined, d) mRNA expression levels of MMP1, MMP2, MMP9, and TMP2 relative to GAPDH are measured by real-time qPCR (Fig.7 - none of the primer set sequences amplify the target gene by UCSC In-Silico PCR analysis, statistical significance of results unknown), and finally potential function of FE and AE extracts at different “Proportion (%)” (?) are devised (context/study parameters unknown).
General comments: Exploration of the marine environment as source material for new medicines is a field of active interest. Algae are a promising bio-source that require further exploration. Given the possibility of rapid isolation and in vitro investigation of bioactive compounds, cell-free as well as cell-based assay offer a convenient screening route. The Manuscript provides partial descriptions of these initial steps as applied to B. leptopoda extracts targeting wound healing, but major as well as minor study deficiencies need to be first addressed.
Major issues: As a basic rule, experimental studies need to provide both accurate as well as sufficiently detailed information to allow study replication by outside labs. Unfortunately, the Manuscript fails to deliver these basic requirements. First, and most importantly, none of the primers amplify the desired mouse targets by in-silico PCR (https://genome.ucsc.edu/cgi-bin/hgPcr). And second, figures lack captions with sufficient detail, including statistical significance, to understand the results.
Minor issues: please see attached reviewed Manuscript. Please address all 33 comments.

Can be improved
Reviewer 3 Report
Comments and Suggestions for Authors
The manuscript by Lin et al. investigates the extraction methanol, ethanol and dd water extracts of Botryocladia leptopoda and its subsequent use in a wound healing application via antioxidant and anti-inflammatory tests. The use of seaweed and its extracted products is a widely studied topic that is highly relevant, especially in today’s global context (as a potential feedstock for climate mitigation). Therefore, this paper deserves recognition in this field of research. The manuscript offers some interesting information, but it still requires some improvement. My major comments are related to the extraction procedure and the reproducibility of this study. For this reason, the reviewer suggests a major revision. My comments are found below.
- Please add some key quantitative results to the abstract section.
- Lines 53-55: is this statement correct? There are numerous (recent) studies on the use of marine macroalgae as a resource for bioactive compounds and their use thereof.
- The introduction section should be extended. Currently, it is written in a very concise and general way. What is the novelty of this manuscript? It is only applying a different macroalgae?
- There are some chemicals missing in section 4.1.
- To what extent were the algae pulverized? What was the particle size (sieve or mesh size used). Particle size determines the extraction efficiency and is therefore important information.
- How were the specific extraction conditions chosen? Why a temperature of 25 °C in case of alkaline and 70 °C/100°C in case of ethanol/ddH2O extraction? Was the reactor content stirred? Please add more details regarding the extraction procedure. How were the extracts separated from the residual biomass? Please include these details.
- Make sure that abbreviations are mentioned in full the first time they appear in the text. This is currently not always the case.
- The authors mention ultrasonication in the materials and methods section. Please add the details? What power/amplitude, frequency,etc., was applied ?
- Line 340: the period should be replaced with a comma.
- Lines 406-408: also depict these as equations as has been done with other equations in the manuscript.
- It is clear that the authors target polyphenols as indicators of potential wound healing activity. This can already be mentioned in the introduction section. Consider also including some recent literature considering polyphenols and their effects on wound healing applications.
- Lines 94-95: please provide a reference.
- In the figures, the bar plots depict error bars. Are these error bars related to the analytical procedures or different extraction experiments? Please elaborate (also include this in the materials and methods section).
- Did the authors perform a (bio-)chemical composition analysis on the original feedstock? Macroalgae composition is highly variable based on different factors. Therefore, biochemical composition analysis is highly advised.
- Sometimes the authors use the term macroalgae, sometimes seaweeds. Be consistent with the use of terminology.
- Figure 3 is currently difficult to read. Suggestion: increase the size of the figures and place the panels below each other.
- Please be consistent with the use of significant figures. Sometimes one decimal, sometimes two decimals are used for the same parameter. Please go through the text (and tables) and adapt.
- The results of the UHPLC method are only very briefly described in the manuscript itself. Is it possible to elaborate a little bit more on this part?
- Line 272: species name should be italicized.
Round 2
Reviewer 1 Report
Comments and Suggestions for Authors
Dear Authors,
Thank you for submitting the revised version of your manuscript entitled “Botryocladia leptopoda Extracts Promote Wound Healing Ability via Antioxidant and Anti-Inflammatory Activities and Regulation of MMP/TIMP Expression.”
I have carefully evaluated the revised manuscript. The scientific content has been substantially improved, and most of my previous concerns have been adequately addressed. The study is interesting and relevant to the field of marine natural products and wound-healing research.
However, I would like to draw your attention to the English language and style, which still require minor polishing before publication. The manuscript is generally clear, but some sentences are overly long, and there are small grammatical issues and inconsistencies in terminology. For example:
- Occasional missing spaces (e.g., “210%and 112%”).
- Typographical errors (e.g., “would healing-related genes” → “wound healing-related genes”).
- Repeated or redundant phrases such as “in this study” and “the results showed that,” which can be simplified.
- Inconsistent capitalization and spacing in extract names (e.g., AE, FE, HE).
Therefore, I recommend that the manuscript undergo a light professional English editing to improve fluency, clarity, and readability.
Aside from these minor linguistic issues, I consider the manuscript scientifically sound and suitable for publication after English revision.
Best regards,
Comments on the Quality of English LanguageThe English in the text needs to be revised.
Reviewer 2 Report
Comments and Suggestions for Authors
REVISED MANUSCRIPT
Comments to the Authors
A revised and improved Manuscript was now submitted that partially addresses concerns raised in the initial peer review. However, no levels of significance are attached to letters ‘a,b,c’ and captions do not identify what comparisons are made.
More concerning are unresolved issues regarding RT-PCR primers. The original MMP-9 RNA-R (5′-TCCAGTACCAAGACAAAGC-3′) primer is now listed as the forward primer, while a completely new reverse primer appears. Worse still, the GAPDH primers do not amplify mouse GAPDH so that MMP-2, MMP-9, TIMP-2, mRNA expression cannot be measured relative to GAPDH by 2-ΔΔCt. These issues seriously undermine any confidence in the reported results even after removal of MMP1 data. Please see the complete primer analysis below for MMP1, MMP2, MMP9, TIMP2, and GAPDH.
ORIGINAL PRIMERS:
MMP-1 RNA-F (5′-GCTAACCTTTGATGCTATAACTACGA-3′)
MMP-1 RNA-R (5′-TTTGTGCGCATGTAGAATCTG-3′)
- GENCODE GENES Amplification result:
- Human: >ENST00000315274.7__MMP1:911+985 75bp GCTAACCTTTGATGCTATAACTACGA TTTGTGCGCATGTAGAATCTG
- Mouse: No matches
ORIGINAL PRIMERS:
MMP-2 RNA-F (5′-ATCGCAGACTCCTGGAATG-3′)
MMP-2 RNA-R (5′-CCAGCCAGTCTGATTTGATG-3′)
- GENCODE GENES Amplification result:
- Human: no matches
- Mouse: >ENSMUST00000211567.1__Mmp2:214+373 160bp ATCGCAGACTCCTGGAATG C CAGCCAGTCTGATTTGATG
ORIGINAL PRIMERS:
MMP-9 RNA-F (5′-CCAGCCAGTCTGATTTGATG-3′)
MMP-9 RNA-R (5′-TCCAGTACCAAGACAAAGC-3′)
- GENCODE GENES Amplification result:
- Human: No matches
- Mouse: No matches
REVISED MANUSCRIPT PRIMERS
MMP9 F: 5’-TCCAGTACCAAGACAAAGC-3’ à Reverse to Forward primer switch
MMP9 R: 5’-GAGCCCTAGTTCAAGGGAC-3’à No relationship to original primer
ORIGINAL PRIMERS:
TIMP-2 RNA-F (5′-GGTAGCCTGTGAATGTTCCT-3′)
TIMP-2 RNA-R (5′-ACGAAAATGCCCTCAGAAG-3′)
- GENCODE GENES Amplification result:
- Human: No matches
- Mouse: >ENSMUST00000017610.9__Timp2:2513+2723 211bp GGTAGCCTGTGAATGTTCCT ACGAAAATGCCCTCAGAAG
ORIGINAL PRIMERS:
GAPDH RNA-F (5′-TGTTGCCATCAATGACCCCTT-3′)
GAPDH RNA-R (5′-CTCCACGACGTACTCAGCG-3′)
- GENCODE GENES Amplification result:
- Human:
>ENST00000920782.1__GAPDH:173+374 202bp TGTTGCCATCAATGACCCCTT CTCCACGACGTACTCAGCG
>ENST00000889647.1__GAPDH:166+367 202bp TGTTGCCATCAATGACCCCTT CTCCACGACGTACTCAGCG
>ENST00000920783.1__GAPDH:166+367 202bp TGTTGCCATCAATGACCCCTT CTCCACGACGTACTCAGCG
>
>
>
…
- Mouse: No matches

No additional comments
Reviewer 3 Report
Comments and Suggestions for Authors
The authors addressed most of the reviewer's comments, and therefore, the reviewer accepts this manuscript for publication in Marine Drugs.
